# Adhesion to Zirconia: A Systematic Review of Surface Pretreatments and Resin Cements

**DOI:** 10.3390/ma14112751

**Published:** 2021-05-22

**Authors:** Rubén Comino-Garayoa, Jesús Peláez, Celia Tobar, Verónica Rodríguez, María Jesús Suárez

**Affiliations:** Department of Conservative Dentristy and Bucofacial Prosthesis, Faculty of Odontology, Complutense University of Madrid, 28040 Madrid, Spain; comino93garayoa@gmail.com (R.C.-G.); celiatobar@gmail.com (C.T.); veranicr@ucm.es (V.R.); mjsuarez@ucm.es (M.J.S.)

**Keywords:** resin bonding, dental bonding, zirconia, 3Y-TZP ceramic, cement

## Abstract

This systematic review aims to evaluate the different pretreatments of the zirconia surface and resin cement in order to determine a valid operative protocol for adhesive cementation. Methodologies conducted for this study followed the Prisma (Preferred Reporting Items for Systematic Reviews and Meta-Analysis) guidelines. An electronic search was performed in four databases. The established focus question was: “What type of surface conditioning method is the one that obtains the best adhesion values to zirconia over time by applying a resin cement?” Forty-five relevant papers were found to qualify for final inclusion. In total, 260 different surface pretreatment methods, mainly combinations of air-abrasion protocols and adhesive promoters, were investigated. Altogether, the use of two artificial aging methods, three types of cement and four testing methods was reported. The results showed that mechanicochemical surface pretreatments offered the best adhesive results. Self-adhesive cement and those containing 10-MDP obtained the best results in adhesion to zirconia. Artificial aging reduced adhesion, so storage in water for 30 days or thermocycling for 5000 cycles is recommended. A standardized adhesive protocol has not been established due to a lack of evidence

## 1. Introduction

In recent decades, the increasing aesthetic needs in dentistry have led to the progressive overcoming of metal-ceramic prostheses and led to a focus on indirect metal-free restorations. Yttrium-stabilized zirconia has occupied an increasingly important role and offers a wide variety of clinical applications, such as root posts, implant abutments or as a material of choice for indirect ceramic restorations. It has the most favorable mechanical properties compared to other high-strength ceramics with flexural strengths of 700–1200 MPa, fracture resistance of more than 2000 N and fracture toughness of 7–10 MPa [1,2,3,4]. However, not only strength is important but also cementation and the adhesion of cement both to the dental tissues and to the restorative material is critical for the long-term success of the restorations [5].

Surface treatment with hydrofluoric acid (HF) and silane coupling agent application of the silica-based ceramics is a well-established method to achieve durable adhesion to resin-based materials [6]. However, this process has failed for adequate resin bond to zirconia ceramics because they do not contain a silica phase making adhesion impossible [7,8,9]. Therefore, in the last few years, several zirconia surface pretreatments have been suggested to enhance the bond strength of luting cement to zirconia ceramics. Some of these methods facilitate an increase of surface roughness, improving micro-mechanical retention of the resin cement employing airborne particle abrasion with alumina particles [10,11], tribochemical silica coating (TSC) [1,12,13]. laser irradiation or chemical etching [14,15,16,17]. However, it has been reported that possible damage on the zirconia surface is created by air-abrasion methods [18,19,20,21]. To solve this problem, alternative methods have been introduced, such as chemical promoters and resin cement based on organophosphate/carboxylic acid monomers specific for zirconia [22] that have been considered as chemical surface treatments. Among them are functional monomers as 10-methacryloyloxydecyl dihydrogenphosphate (10-MDP), phosphonic acid acrylate or anhydrides [23,24,25]. Furthermore, silane deposition [26], selective infiltration etching (SIE) [27], ceramic coating and the use of cement-containing MDP are proposed chemical methods [28]. However, hydrolytic degradation is still problematic [29].

Several methods have been used to evaluate the bond strength of resin-based materials to dental ceramics, including macroshear, microshear, macrotensile and microtensile tests. Furthermore, methods to evaluate bond durability simulating the oral conditions include short- and long-term water storage and thermocycling at diverse temperatures, dwell time, and number of cycles. Therefore, it is difficult to compare different studies on the same materials even when the same test method was employed [5,30].

Due to the great increase in in vitro studies in recent years and the lack of consensus on resin-bonding protocols for zirconia restorations, it is necessary to evaluate the current data to unify criteria and provide clinicians with relevant information for their daily activity. Therefore, the aim of this study was to systematically review the literature to evaluate the different zirconia surface pretreatments and resin cement to determine a valid clinical protocol for adhesive cementation.

## 2. Materials and Methods

### 2.1. Search Strategy

The present systematic review followed the Preferred Reporting Items for Systematic Reviews and Meta-Analyses (PRISMA) guidelines [31]. The practice-orientated research/focused question was: “What type of surface conditioning method is the one that obtains the best adhesion values to zirconium over time by applying a resin cement?”

The literature search was performed by two independent reviewers (R.C.-G.; C.T.), including articles published between 1 April 2015 and 1 December 2020, and any disagreement was resolved by a third reviewer (M.J.S.). The electronic databases screened were PubMed, The Cochrane Library, Scopus, and Web of Science (WOS). The following search terms and their combinations were employed: “zirconium,” “zirconia,” “3Y-TZP,” “3Y-TZP ceramic,” “dental adhesion,” “dental bonding,” “bond strength test,” “cement” and “resin bonding.”

### 2.2. Elegibility Criteria

The articles included followed the inclusion criteria: published in English, in vitro studies that reported on adhesion to zirconia restorations by resin cement, using microtensile, macrotensile, microshear or macroshear bond tests with results (mean and standard deviation) in MPa and subjected to artificial aging. Studies with the following criteria were excluded: studies on high translucent zirconia, data were not presented in MPa or without normal distribution, studies with a number of specimens less than 5, studies with samples subjected to less than 5000 thermocycles (TC), or less than 1 month of storage. Clinical trials, case reports, case series and pilot studies were also excluded. Studies that did not specify any of these data were removed from the review. Any disagreement regarding the eligibility of the included studies was resolved through discussion and consensus. The inclusion and exclusion criteria are listed in Table 1.

### 2.3. Data Extraction and Collection

The following variables were extracted: mean and standard deviation of the bond strength recorded in MPa, type of pre-treatment technique, type of cement, methods of aging (storage conditions and duration and/or the number of thermocycling procedures) and type of bond strength test.

## 3. Results

The research carried out in PubMed (Table 2), Cochrane (Table 3), Scopus (Table 4) and WOS (Table 5) identified 158, 13, 117 and 640 studies, respectively. Search results are presented graphically in Figure 1. Of the total of 928 articles, 219 were discarded as duplicates, obtaining a total number of 709 studies. After reading the abstract, a further 614 were eliminated because the information was not relevant, they were not in vitro studies, or no artificial aging was conducted. The full texts of the 95 remaining studies were read, and a further 50 were discarded as they failed to fulfill the inclusion criteria. The most common reasons for rejection were the lack of data in the methodology regarding the bond strength, or aging conditions were <1-month storage or TC < 5000. After this screening, a final total of 45 studies were included in the systematic review. The included articles were from 12 countries, spanning North America, South America, Central Europe, North Africa, and Asia. The selection processes as a PRISMA flow diagram were summarized in Figure 2.

In the selected 45 articles, a total of 511 experimental groups were identified with bond strength results (MPa). In these groups, the search identified 260 different methods of zirconia surface pretreatments prior to adhesive cementation. Of the selected 45 articles, 43 articles did not report a control group with no conditioning surface. In the other two articles, two experimental groups were identified as control groups. Pretreatment techniques were listed as described by the authors and classified into three groups: Mechanical conditioning methods using abrasives were found in 140 experimental groups; chemical methods using adhesive promoters were used in 5 experimental groups; mechanicochemical conditioning methods based on using abrasives or etchants followed by adhesion promoters were practiced in 364 experimental groups. The main zirconia pretreatments are summarized in Figure 3.

Resin cement is summarized in Figure 4. Table 6 displays the resin cement used that were listed and categorized according to their main chemical compositions. Three types of resin cement were identified in the 511 experimental groups: Bisphenol A-glycidyl methacrylate (Bis-GMA) (n = 78), MDP (n = 251) and self-adhesive (n = 182).

Figure 5 summarize four testing methods identified to evaluate adhesion of resin cement to zirconia in the 511 experimental groups: macrotensile (n = 44), microtensile (n = 137), macroshear (n = 301) and microshear (n = 29). The mean bond strength ranged from 0 to 47.1 MPa.

Regarding the aging conditions, the type and duration of aging methods were recorded. Figure 6 summarizes the methods identified for artificial aging. Specimens were aged by thermocycling in 35 of the 45 articles selected. In total, 205 groups were not thermocycled, and 306 experimental groups were thermocycled at varying number of cycles ranging between 5000 and 37,500. The storage media showed a great variation: tap water, distilled water, saliva, phosphoric acid, alcohol, sodium hydroxide, acetic acid, esterase, and acidic and alkaline solutions. Storage duration ranged from 30 to 1825 days.

## 4. Discussion

In the present systematic review, the studies were conducted in vitro, and a great heterogeneity was observed among the studies. Despite the limitations of this type of study, it is necessary to evaluate the behavior of different materials and techniques before their clinical application [18,32,33].

### 4.1. Zirconia Surface Pretreatments

In this systematic review, pretreatment techniques were classified into three groups: (1) mechanical: studies that used air-abrasion protocols, laser, ceramic coating, or chemical etching, (2) chemical: studies that employed coupling agents such as adhesive resins, silanes or primers, (3) mechanicochemical: when both mechanical and chemical conditioning methods were applied. Control groups were defined as zirconia substrates with no surface pretreatment.

The studies agree that the zirconia surface needs to be prepared before applying the resin cement since all the pretreatments increased the bond strength, improving the values of the control group [34,35,36]. The first requirement for adhesion is to achieve a surface free of contaminants. Most of the studies started the surface conditioning protocol by polishing with papers, sprays or milling cutters of silicon carbide ranging between 220 to 4000 grit. Although several studies did not mention this step, they may have done it too. Ultrasonic cleaning before surface conditioning or the resin cement is also widely used [32,36,37,38,39,40,41,42,43,44,45,46,47,48,49]. Likewise, several solutions were used, including distilled water, alcohol, acetone, ethanol, and isopropanol, with a usage time between 1 and 10 min. In almost no studies, the effect of cleaning methods on adhesion to zirconia has been considered, but all authors considered it as a beneficial element [37,42,44,48].

Several mechanical pretreatments have been investigated. Sandblasting with alumina particles improved the bond strength values due to the increase in surface energy, wettability, roughness, and the appearance of hydroxyl groups that will facilitate bonding with the primer/universal adhesive/cement [34,38,41,43,48]. The particle size used ranged from 30 to 110 µm, at 0.5–4 bar for 10–20 mm [30,33,34,35,36,38,39,40,41,43,44,45,46,48,49,50,51,52,53,54,55,56,57,58,59,60,61,62,63,64,65,66,67,68,69,70,71,72]. An increase in particle size and pressure had long been associated with the formation of microcracks and weakening the mechanical properties of the material [35,39,41,49,58,61,62,65,67,68,70]. However, the bond strength was not affected by the variation in particle size and pressure [48]. It has also been reported that sandblasting before sintering caused fewer phase transformations than after sintering. However, sandblasting before or after sintering had no influence on adhesion [48,67].

The application of lasers to the surface of zirconia is based on the same principle as sandblasting, obtaining a rough surface and an increase in wettability that allows micromechanical retention with the resin [44]. Different types of lasers have been described (Er: YAG, Nd: YAG, Yb: YAG, CO_2_), with different parameters of power, energy intensity, distance, and duration. Most of the studies concluded that the application of laser did not increase the bond strength compared to sandblasting and did not obtain acceptable adhesion values [36,40,43,51] due to the appearance of microcracks on the surface of the zirconia, leading to a phase transformation and weakening the mechanical properties [51]. Therefore, the laser is not currently considered a valid mechanical pretreatment [36,43].

An electrical discharge machine (EDM) described by Rubeling et al. [73] was used in one study, obtaining better adhesion values than sandblasting and TSC, but the presence of microcracks was also seen on the surface of zirconia [39].

Recently, plasma has been introduced as zirconia surface pretreatment to increase the surface energy and optimize the chemical surfaces of the substrates without affecting its structural properties. However, the application of oxygen or argon plasma did not obtain good adhesion values after artificial aging, which added to the appearance of impurities on the surface of the zirconia, indicated its susceptibility to hydrolytic degradation [49,55,71].

Even though zirconia is a polycrystalline ceramic that cannot be etched due to the absence of a silica phase, different etching methods have been tested in acidic solutions such as phosphoric, nitric and HF acid. These solutions showed contradictory results compared to sandblasting [56,61,63,68]. In addition, their negative effects must be evaluated to be able to propose this surface conditioner method as safe and effective [63,68].

Several ceramic coating methods have been also investigated. It has been reported that the application of a layer of silica glaze creates a more reactive and etchable glass surface that can be treated as a glass-ceramic. The posterior HF etching removes the glassy matrix creating a porous surface with high surface energy, ideal for cement penetration [33]. This pretreatment showed high bond strength before and after thermocycling [33]. Fusion sputtering described by Aboushelib in 2012 [74] creates a rough surface on zirconia through the spraying of microscopic zirconia particles on non-sintered zirconia that fused structurally with zirconia after sintering, increasing its surface. This method obtained better adhesion values compared to sandblasting [54]. Likewise, nanostructured alumina coating in aluminum nitrate improved the bond strength compared to sandblasting and TSC due to the appearance of a rough surface [70]. Dos Santos et al. [75] incorporated titanium dioxide nanotubes onto the zirconia surface before sintering to increase the surface energy; however, they did not have a significant effect on adhesion. SIE was also used to modify the zirconia surface. Zirconia is coated with silica-based material that diffuses in the zirconia structure during the fusion at 960 °C, followed by the application of HF for 10 min to dissolve the glass component. This method obtained contradictory results, and more studies are needed to demonstrate its efficacy [33,35].

To simplify the steps in the adhesion and make life easier for clinicians, the adhesives started to contain chemical promoters, being called universal. The objective was to achieve the preparation of the restoration without the need to add another component. Most of these universal adhesives contain 10-MDP at different concentrations and have been the most studied products in the last five years. It has been reported that the application of a universal adhesive with 10-MDP increased adhesion to zirconia after sandblasting [34,38,45,49,50,55,69] and has even been proposed to replace mechanical conditioning and the application of a primer [38,57]. The photopolymerization or not of the adhesive before the application of cement had no relevance on adhesion [76]. However, the main problem of 10-MDP is hydrolytic degradation, which causes a decrease in adhesion over time in all its application forms, compromising the adhesive protocol [50,65,66,69].

In the reviewed articles, most of the studies involved mechanicochemical methods [30,32,33,34,35,36,37,38,39,40,41,42,43,44,45,46,47,49,51,52,53,55,59,60,61,63,66,72,75,76,77]. Although sandblasting can modify the surface of the zirconia, when used alone, it has been shown to be ineffective in increasing adhesion to zirconia, and a chemical surface conditioner is required to make it stable in the long term [33,34,51,70]. This is consistent with previous systematic reviews [5,7,78]. These chemical conditioners contain various molecules found in primers, adhesives, or cement. Certain chemical promoters based on organophosphate monomers such as 10-MDP, 6-methacryloyloxyhexyl phosphonoacetate (6-MHPA) or 4-methacryloyloxyethy trimellitate anhydride (4-META) have demonstrated to increase the bond strength after sandblasting [59,64]. Currently, MDP is the most widely used, and it has been reported that primers without 10-MDP showed lower bond strength values [41,48]. However, other studies showed that the use of primers with 10-MDP after sandblasting did not improve adhesion and were not consistent over time, but they were better than other primers without this molecule [65,69].

TSC consists of zirconia sandblasting with silica-coated alumina particles. The impact of the particles creates an irregular surface incorporating silica into the zirconia structure and allowing the use of silane as a binding agent to silica and resin. This leads to the appearance of chemical chains of siloxane between cement and residual silica, increasing adhesion and improving the wettability and surface energy of zirconia [34,76]. The particle size ranged from 30 to 110 µm, at 0.8–4 bar and at 10 mm. However, cracks occurred on the surface of zirconia at high pressure, and a pressure of 1.8–2.8 bar has been proved to be sufficient to achieve a significant increase in adhesion [32,40]. The TSC showed better bond strength than conventional sandblasting, favoring long-term stable adhesion [34,39,66,70,76]. However, it is not yet clear whether the application of a primer with silane and 10-MDP has a beneficial effect compared to the application of silane alone [70,76]. Some studies advised that TSC should be performed after sintering the zirconia and did not recommend cleaning with water or ultrasonically prior to silane application [42,67]. This procedure had been considered an alternative to conventional sandblasting together with the application of 10-MDP [36], although the two surface pretreatments are equally valid [37].

Feldspathic ceramic sandblasting and silane application appears to be a promising surface pretreatment, showing statistically better values than the TSC plus silane after thermocycling [7]. Thammajaruk et al. showed that this surface pretreatment applied to a 10-MDP primer obtained the best long-term values [79]. One study showed that the application of Yb: YAG laser combined with silane was better than sandblasting or TSC combined with silane, both before and after aging [44].

Other methods to silicatize the zirconia surface have been introduced. The Silano-Pen system consists of a lighter with a solution of butane and silane. When butane is burned, the silane compound decomposes into SiOx-C fragments that adhere to zirconia, allowing it to be silanized. However, its effectiveness has not been proved compared to other methods [40].

### 4.2. Resin Cements

The classification of resin cement was complicated because of the great variation in their chemical compositions: phosphoric acid esters, 10-MDP, HEMA, glycerolphosphate dimethacrylate (GPDM), 4-META, bis-GMA or triethylene glycol dimethacrylate (TEGDMA). In addition, the exact composition or percentage of each component is hardly shown due to the lack of information from manufacturers. Therefore, their classification was structured in self-adhesive, cement with 10-MDP, and Bis-GMA cement (without 10-MDP or were not self-adhesive). In general, within the same group, the cement had great variability due to both the percentage of the different components and the viscosity of the cement, which can interfere with micromechanical interpenetration [60]. There is no consensus on which cement is above another, except for Bis-GMA, which showed lower adhesion values than the other two groups. However, this molecule better withstands hydrolytic degradation [34,60]. The relationship to the addition of a primer containing 10-MDP is unclear. Different studies have reported an increase in adhesion when previously applying a 10-MDP primer, especially with self-adhesive cement [41,52,53]. Conversely, another study reported the opposite in cement with 10-MDP due to the saturation of this molecule [53]. Nevertheless, there is consensus on the need for previous mechanical surface conditioning to increase their adhesive values [51,58,60,70]. Regarding the degradation of cement after artificial aging, no consensus exists. Thus, more studies are needed to demonstrate the ideal resin cement [30,60].

### 4.3. Test

Different types of tests have been used to assess the bond strength between zirconia and composite cement that can be explained by the lack of an international standard. The most used was the macroshear test, probably due to its simplicity of use. Otani et al. [80] described the macro tests (macroshear and macrotensile) as those that presented more heterogeneity in the distribution of stress and loads due to the greater adhesion surface. On the other hand, the micro tests (microshear and microtensile) showed less variation and higher adhesive values due to a smaller adhesion area and less possibility of finding defects in the cementing. However, the number of premature failures in the specimen preparation step was higher. Nevertheless, the variability of the tests and their influence on the results make it very difficult to compare the results among the studies.

### 4.4. Artificial Aging

The most used method for artificial aging was liquid storage and thermocycling. Liquid storage allows the evaluation of hydrolytic degradation, and thermocycling reproduces in vitro hydrothermal aging [5,78]. The most frequently used liquid was distilled/deionized water, but other types of solutions were used, such as esterase, acetic acid, alcohol, phosphoric acid or artificial saliva, to reproduce different clinical scenarios [35,46,62]. Studies concluded that storage in a liquid medium significantly reduced adhesion compared to control groups. Acetic acid, phosphoric acid and esterase were the solutions that caused a greater effect [32,35,42,46,64,65]. The number of cycles showed a great variation among the studies with thermocycled groups, which makes it impossible to compare the results. In this systematic review, the ISO 10477 standard was followed concerning metal–resin bond, which established the minimum number of cycles at 5000 [81]. Thermocycling decreased adhesion values due to hydrothermal aging [47,51,52,68]. However, it has been reported that the number of cycles above 5000 does not decrease the values significantly [35,47,77]. Other studies used a combination of storage in liquid medium and thermocycling, which caused a significant decrease in the adhesive values [58,64,67]. This combination may be the one that causes greater degradation at the interface but requires much more time to complete [58,64,67].

Since this systematic review was based on in vitro articles, it was not possible to perform clinical guidelines because there are certain clinical factors, such as saliva contamination or parafunctional habits, that negatively affect the adhesion [78]. Furthermore, contradictory results have been found in in vitro studies due to the heterogeneity of study designs that do not provide sufficient evidence to support the selection of a specific technique for better adhesion. Further studies are necessary to evaluate promising surface pretreatments techniques, as well as clinical trials to be able to indicate a clinical protocol with predictable results. In addition, due to the current boom in highly translucent zirconia, it would be advisable to carry out a new systematic review in a few years trying to establish an adequate surface conditioning for the adhesion of resin cement and to compare it with traditional zirconia. Moreover, certain recommendations must be considered for future studies and reviews: (1) Studies should include a control group with no treatment to more effectively assess the pretreatment tested. (2) Two types of tests should be performed within each study to avoid variability in the results obtained. (3) Manufacturers and authors must provide the composition of primer, universal adhesive and cement used. (4) It is necessary to standardize the artificial aging method used to compare the results in a more effective way.

## 5. Conclusions

Within the limitations of this review, the following conclusions were drawn:-There are a great variety of zirconia surface pretreatments, cement, artificial aging method and tests used in the studies that make it difficult to compare the results.-Zirconia surface cleaning must be performed before pretreatment methods to adhesion.-Mechanicochemical surface pretreatments offered the best adhesive results. Tribochemical silica coating at a pressure of 1.8–2.8 bar has proved to achieve a significant increase in adhesion to zirconia.-New methods as feldspathic ceramic sandblasting and silane application or YAG laser combined with silane seem to be promising alternatives in adhesion to zirconia.-There is great variability in the percentage of components and the viscosity of the resin cement. Self-adhesive cement and those containing 10-MDP obtained the best results in adhering to zirconia, without clarification of which is the best.-The use of a 10-MDP primer is still controversial.-Standardization of test to evaluate the bond strength between zirconia and resin cement is needed-Artificial aging decreased adhesion; therefore, storage in water for 30 days or thermocycling for 5000 cycles must be performed in laboratory studies.-A clinical protocol for adhesive cementation to zirconia has not yet been performed.

## Figures and Tables

**Figure 1 materials-14-02751-f001:**
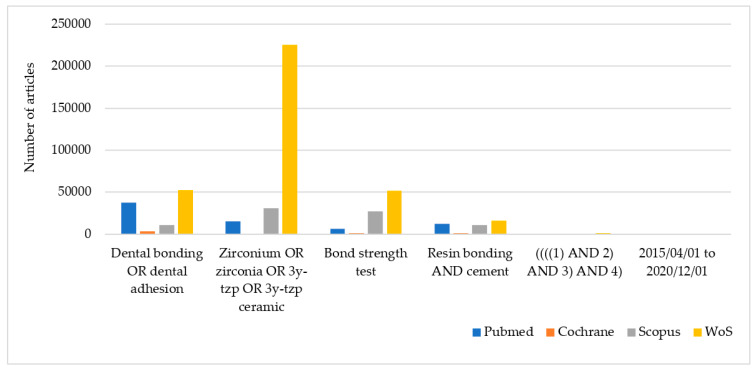
Research results by databases.

**Figure 2 materials-14-02751-f002:**
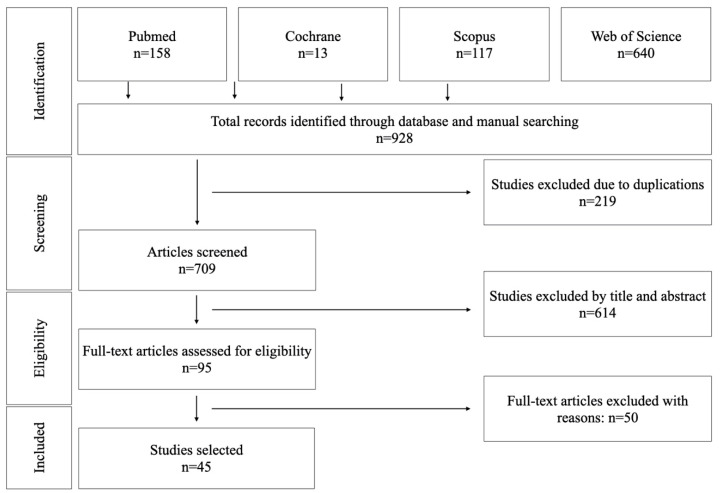
Flow chart of the studies selection process.

**Figure 3 materials-14-02751-f003:**
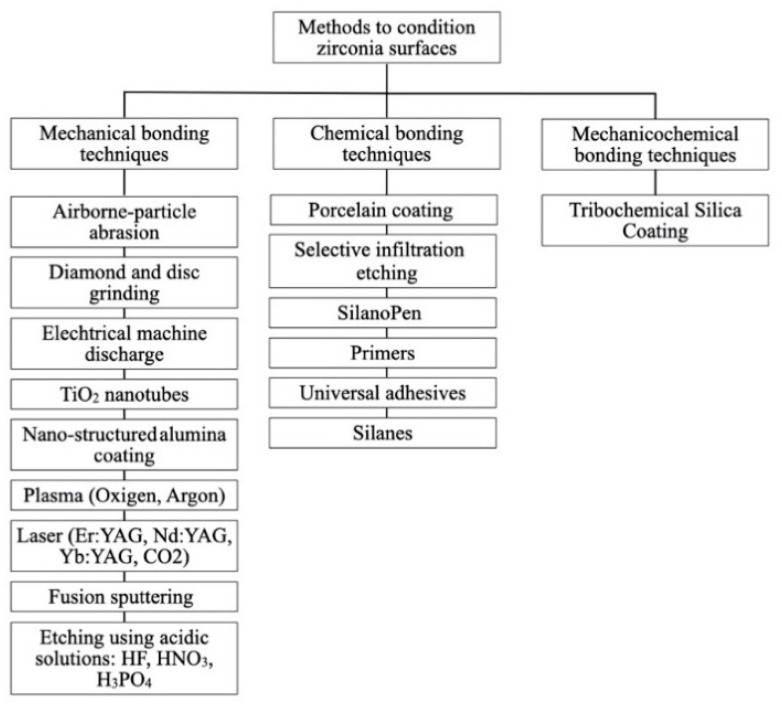
Methods to condition zirconia surfaces.

**Figure 4 materials-14-02751-f004:**
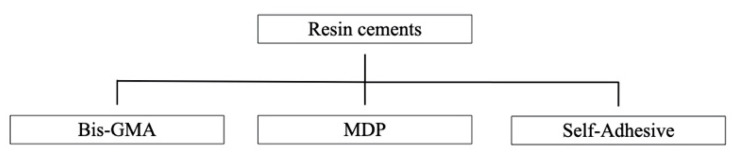
Resin cement.

**Figure 5 materials-14-02751-f005:**
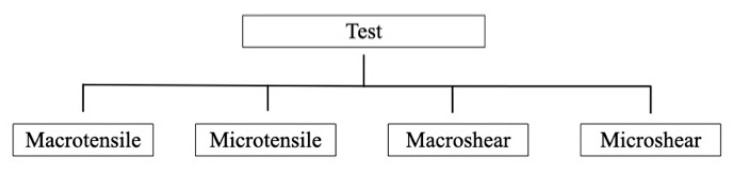
Bond strength testing methods.

**Figure 6 materials-14-02751-f006:**
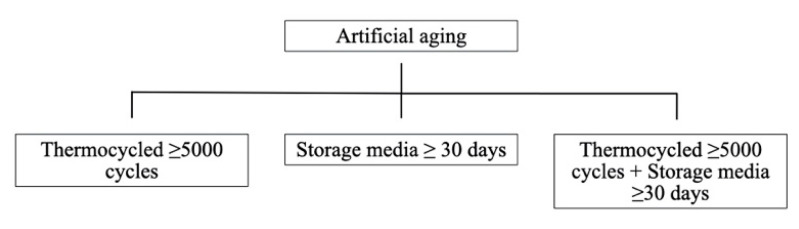
Artificial aging methods.

**Table 1 materials-14-02751-t001:** Inclusion and exclusion criteria.

**Database**	Pubmed; Cochrane Library; Scopus; Web of Science
**Publication Date**	1 April 2015–1 December 2020
**Keywords**	“dental bonding,” “dental adhesion,” “zirconium,” “zirconia,” “3Y-TZP,” “3Y-TZP ceramic,” “bond strength test,” “cement,” and “resin bonding.”
**Language**	English
**Type of Paper**	In vitro studies
**Inclusion** **Criteria**	Studies evaluating adhesion between zirconia and resin cement, studies performed with microtensile, macrotensile, microshear or macroshear test and artificial aging
**Exclusion** **Criteria**	Translucent zirconia, absence of bonding strength evaluation, data not presented in MPa or without normal distribution, number of specimens <5, insufficient aging (TC < 5000 or storage < one month), studies performed with pull-out test, clinical trials, case reports, case series and pilot studies
**Journal** **Category**	All

**Table 2 materials-14-02751-t002:** Pubmed research.

Search	Literature Search Strategy	Results
1	“dental bonding” OR “dental adhesion”	37,888
2	“zirconium” OR “zirconia” OR “3y-tzp” OR “3y-tzp ceramic”	15,396
3	“bond strength test”	6663
4	“resin bonding” AND “cement”	12,418
5	((((1) AND 2) AND 3) AND 4)	414
6	Filters: Publication date from 1 April 2015–1 December 2020	158

**Table 3 materials-14-02751-t003:** Cochrane research.

Search	Literature Search Strategy	Results
1	“dental bonding” OR “dental adhesion”	3178
2	“zirconium” OR “zirconia” OR “3y-tzp” OR “3y-tzp ceramic”	817
3	“bond strength test”	999
4	“resin bonding” AND “cement”	1524
5	((((1) AND 2) AND 3) AND 4)	42
6	Filters: Publication date from 1 April 2015–1 December 2020	13

**Table 4 materials-14-02751-t004:** Scopus research.

Search	Literature Search Strategy	Results
1	“dental bonding” OR “dental adhesion”	10,936
2	“zirconium” OR “zirconia” OR “3y-tzp” OR “3y-tzp ceramic”	30,821
3	“bond strength test”	26,888
4	“resin bonding” AND “cement”	10,892
5	((((1) AND 2) AND 3) AND 4)	253
6	Filters: Publication date from 1 April 2015–1 December 2020	117

**Table 5 materials-14-02751-t005:** Web of Science research.

Search	Literature Search Strategy	Results
1	“dental bonding” OR “dental adhesion”	52,377
2	“zirconium” OR “zirconia” OR “3y-tzp” OR “3y-tzp ceramic”	225,964
3	“bond strength test”	51,817
4	“resin bonding” AND “cement”	16,345
5	((((1) AND 2) AND 3) AND 4)	1249
6	Filters: Publication date from 1 April 2015–1 December 2020	640

**Table 6 materials-14-02751-t006:** Resin cement studied according to their main composition.

Type	Cement	Manufacturer
Bis-GMA	BiFix	VOCO
	BiFix QM	VOCO
	Calibra Esthetic	Dentsply Sirona
	Clearpearl DC	Kuraray Noritake
	RelyX Veneer	3M ESPE
	Variolink II	Ivoclar Vivadent
	Variolink Esthetic DC	Ivoclar Vivadent
	Variolink N	Ivoclar Vivadent
MDP	Clearfil SA	Kuraray Noritake
	Multilink Speed	Ivoclar Vivadent
	Panavia 21	Kuraray Noritake
	Panavia F	Kuraray Noritake
	Panavia F2.0	Kuraray Noritake
	Panavia SA	Kuraray Noritake
	Panavia V5	Kuraray Noritake
	Permacem	DMG
	Permacem 2.0	DMG
	Theracem	BISCO
Self-adhesive	BiFix SE	VOCO
	DuoCem	Coltene
	Duo-Link Universal	BISCO
	G-Cem Link Ace	GC
	Maxcem Elite	Kerr Dental
	Multilink Automix	Ivoclar Vivadent
	NX3	Kerr Dental
	RelyX ARC	3M ESPE
	RelyX U100	3M ESPE
	RelyX U200	3M ESPE
	RelyX Ultimate	3M ESPE
	RelyX Unicem	3M ESPE
	RelyX Unicem 2	3M ESPE
	ResiCem	Shofu Dental
	SeT PP	SDI
	SmartCem 2	Dentsply Sirona

## Data Availability

No new data were created or analyzed in this study. Data sharing is not applicable.

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
