# Peer review of "Adhesion to Zirconia: A Systematic Review of Surface Pretreatments and Resin Cements"

_materials, 2021, doi:10.3390/ma14112751_

Round 1
Reviewer 1 Report
It is not clear why the search criteria in tables 2-5 are not the same. They should also be ordered in the same way and results presented graphically.
Table 6 is not very useful for readers and maybe should be removed from the paper.
Figure 2 is very instructive and authors should prepare similar figures for each subchapter in the chapter 4.
Reviewer 2 Report
The authors deal with the review of the surface pretreatment process and resin cement for dental purposes. The review is well prepared, the authors carefully describe how the review was prepared. The authors used few various databases: PubMed, Cochrane, Scopus, WoS. However, the authors should include also the Google scholar database.
The authors should seriously improve their conclusions. The review includes 82 references, in this case the conclusions should be longer.
Reviewer 3 Report
Title: Adhesion to zirconia: A systematic review of surface pretreatments and resin cements.
1) The subject is relevant and interesting. The study development required to Authors care and persistence. They were also be conditioned by a wide typology of studies and no fewer methodologies.
2) M&M
a. The inclusion and exclusion criteria section has to be accurately defined.
In particular the Authors extracted mean and standard deviation. So, it is expected that only studies showing normal distribution (i.e. where mean and standard deviation are representative of the sample) were included. If so, it has to be specified, if otherwise it has to be clarified on which different sort of data (not mean and standard deviation) the Authors drew their conclusions for this study.
The exclusion criteria reported are inaccurate. A frame in figure1 reports beside "removed due to exclusion criteria", the sentences "lack of data in the methodology" and " study objective not in accordance". The last two has to be qualify. In the section inclusion and exclusion criteria has to be declared what is meant by "exhaustive methodology" and "study objective not in accordance". Otherwise the study would be difficult to read.
b. From 116 to 126 line, in particular, but also elsewhere there is a confusion between group of treatment and group of samples. It has to be conveniently amended.
3)
a) Pretreatment technique was not mentioned completely in several studies (line: 153, 164). So, were these "incomplete" studies considered in the present study or not?
b) The Authors reported that an increase in particle size and pressure had long been associated with the formation of microcracks causing a phase transformation in zirconia from tetragonal to monoclinic and weakening the mechanical properties of the material. The sentence is confused and it is hard to understand if the decreased mechanical properties were due to microcracks or phase transformation or both alterations. The monoclinic and the tetragonal crystal system are two of the seven crystal systems. The monoclinic is subdivisible in three and the tetragonal in seven point groups, and much more besides. The too energy sandblasting could cause microcracks, but about phase transformation either it is convincing explained or it is better to avoid confusions.
